# Research on Intelligent Monitoring of Boring Bar Vibration State Based on Shuffle-BiLSTM

**DOI:** 10.3390/s23136123

**Published:** 2023-07-03

**Authors:** Qiang Liu, Dingkun Li, Jing Ma, Zhengyan Bai, Jiaqi Liu

**Affiliations:** 1Key Laboratory of Advanced Manufacturing and Intelligent Technology, Ministry of Education, Harbin 150080, Chinabzy98098@163.com (Z.B.);; 2Postdoctoral Research Station of Electrical Engineering, Harbin University of Science and Technology, Harbin 150080, China

**Keywords:** boring bar vibration, condition monitoring, deep learning, signal processing

## Abstract

Due to its low stiffness, the boring bar used in deep-hole-boring is prone to violent vibration during the cutting process. It is often inaccurate and inefficient to judge the vibration state of the boring bar through artificial experience. To detect the change of the vibration state of the boring bar over time, guide the adjustment of the processing parameters, and avoid wastage of the workpiece and the loss of equipment, it is particularly important to intelligently monitor the vibration state of the boring bar during processing. In this paper, the boring bar is taken as the research object, and an intelligent monitoring technology of the boring bar’s vibration state based on deep learning is proposed. Based on grouping convolution, channel shuffle, and BiLSTM, a shuffle-BiLSTM NET model is constructed, which is both lightweight and has a high classification accuracy. The boring experiment platform is built, and 192 groups of cutting experiments are carried out. The three-way acceleration and sound pressure signals are collected, and the signals are processed by smoothed pseudo-Wigner–Ville distribution. The original signals are transformed into a 256 × 256 × 3 matrix obtained by a two-dimensional time–frequency spectrum diagram. The matrix is input into the model to recognize the boring bar’s vibration state. The final classification accuracy is 91.2%. A variety of typical deep learning models are introduced for performance comparison, which proves the superiority of the models and methods used in this paper.

## 1. Introduction

The intelligence of cutting is of great significance to the intelligence of the whole manufacturing industry [1]. Intelligent monitoring in machining is a key technical link to realize intelligent manufacturing. Intelligent condition monitoring can realize real-time perception of the state of the equipment in the processing process, guide the adjustment of processing parameters, optimize the product quality, and give a timely warning when the equipment life is insufficient or failure occurs. In the field of mechanical processing, hole processing accounts for one-third of the total, and deep-hole processing accounts for more than half of the hole processing [2]. Due to the particularity of the deep-hole parts structure, a boring bar with a large aspect ratio is generally needed for processing. This kind of boring bar is generally weak in rigidity, and it can easily violently vibrate or even chatter during processing. In the cutting process, the occurrence of vibration and chatter will affect the surface quality of the machined surface, accelerate tool wear, and even cause chatter marks on the surface of the workpiece and damage the workpiece and tools [3]. In deep-hole machining, it is generally difficult to directly observe the vibration state of the boring bar, so the state monitoring technology combined with deep learning can play a key role in vibration monitoring. Timely detection of the change of the vibration state of the boring bar can help guide the adjustment of processing parameters and improve the quality and efficiency of deep-hole processing.

Artificial intelligence technology is an important tool to realize the intelligence of cutting processing, and deep learning is an important part of artificial intelligence technology. Deep learning is derived from machine learning. Through the ability of adaptive feature extraction and learning of multi-layer neural networks, the recognition, classification, and regression of data or images are realized. In recent years, with the advancement of computer technology, deep learning has also rapidly developed and is widely used in the field of machining [4]. In order to give full play to the ability of deep learning to adaptively extract features, some scholars choose to input the original signal of the sensor into the deep learning model. Li et al. [5] input the vibration and sound signals in the boring process into LSTM to realize the state recognition of deep-hole-boring tool bluntness and tool breakage. Liu et al. [6] used the PHM2010 dataset as the training data, used the parallel residual network to adaptively extract the internal features of the multi-dimensional sensor signal, used the stacked bidirectional long short-term memory network to extract the time series features in the signal, and then established an effective mapping with the tool wear value, and a deep learning model with high accuracy was successfully constructed. He et al. [7] constructed a long short-term memory neural network, and input the force, vibration, and acoustic emission signals in the milling process into the network model. Finally, an effective mapping between data and tool wear values was established, which proved the effectiveness and feasibility of the network. Xu et al. [8] designed a deep neural network that combines one-dimensional dilated convolution kernels with residual blocks and used this model to predict the wear of tap tools.

Although deep learning can adaptively extract data features, in many cases, the features extracted by the model itself do not fully reflect the characteristics of the data. Therefore, more researchers consider using the data processed by mathematical methods as the input of the deep learning model for training. Zhou et al. [9] decomposed the spindle torque signal by EMD and input the feature matrix composed of process parameters and workpiece information into the LSTM model to realize the prediction of the tool life. Chen et al. [10] combined CNN with a deep bidirectional gated recurrent unit neural network, collected the milling acceleration signal, input the signal into the neural network after wavelet threshold denoising, and introduced the attention mechanism to adaptively perceive the network weight associated with the wear state, so as to realize real-time and accurate prediction of the tool wear state. Li et al. [11] collected the spindle current signal of the machine tool, used compressed sensing to compress the frequency domain characteristics of the signal, and input the data into the stacked sparse auto-encoder network for training, which successfully realized the recognition of the wear state of the milling cutter.

Considering that deep learning is widely used in the field of image recognition, in order to make better use of the ability of the convolution kernel in the deep learning model to extract image features, some scholars also process data into images as the input of the model. Ren et al. [12] proposed the method of the spectral principal energy vector to combine eigenvalues into a 64 × 64 feature map, an 8-layer CNN network was designed to predict the bearing life, and the smoothing technique was used to solve the problem of discontinuous prediction results. Wen et al. [13] improved the LeNet-5 model by using the motor-bearing dataset, the self-priming centrifugal pump dataset, and the axial piston hydraulic pump dataset, and converted the original signal into a two-dimensional image as data input to verify the fault identification accuracy of the model. Liu et al. [14] proposed a milling chatter monitoring method based on unlabeled dynamic signals. It uses the unsupervised clustering algorithm, does not need to add labels to the data, satisfies any processing parameters and processing environments, has strong stability, and can effectively identify chatter. Pagani et al. [15] processed RGB and HSV channel images of chips as input data to predict tool wear, and the method was used in stable processing scenarios.

It can be seen from the above literature that data processing and deep learning technology have been widely used in the field of cutting. Scholars in various countries have conducted in-depth research on the identification and monitoring of the tool life and equipment status. However, there are few research results on real-time intelligent monitoring of boring bar vibration by deep learning. To accurately and real-time monitor the vibration state of the boring bar during machining, a monitoring method of the boring bar vibration state based on Wigner–Ville distribution and the shuffle-BiLSTM network is proposed by combining data time–frequency analysis technology with deep learning image recognition ability. Based on SPWVD, the time–frequency domain features are extracted from the collected signal and used as the input of the model, which solves the problem that the features extracted from the signal by the deep learning model cannot fully reflect the data characteristics. The shuffle unit is used in combination with BiLSTM. The memory ability of the network is enhanced, and the recognition accuracy is effectively improved through the BiLSTM structure. The training time of the model is shortened, and the real-time monitoring is improved by use of the shuffling unit, with the characteristics of lightweight and high speed. Through the experimental analysis, it is proven that the model runs fast and has a high recognition accuracy, and it has good research and application value.

## 2. Related Work

### 2.1. Time–Frequency Domain Feature Extraction of Sensor Signal Based on SPWVD

#### 2.1.1. Wigner–Ville Distribution

The general linear time–frequency analysis method is not intuitive enough to describe the signal characteristics, such as STFT. When calculating, it is necessary to add a window to the signal, and if a higher resolution of the frequency transform is required, the required window function length will be longer [16]. According to the uncertainty principle, the window function cannot increase or decrease the time and frequency resolutions at the same time. If one of the two increases, the other will decrease [17]. In this case, the trade-off between time and frequency cannot be well-achieved. Therefore, the quadratic time–frequency representation is introduced. Since no window function is added, it avoids the defects of linear time–frequency representation and is a more intuitive time–frequency representation method.

The Wigner–Ville distribution (WVD) can represent the energy of the original signal in both the time domain and the frequency domain. It is a time–frequency analysis method with practical physical significance. Let the original real signal be s(t), then, Hilbert transform is applied to the signal, and its analytical signal a(t) is obtained, as shown in Formula (1):(1)$$a(t)=H[s(t)]=h(t)*s(t)=\int_{-\infty }^{\infty }s(\tau )h(t-\tau )d\tau =\frac{1}{\pi }\int_{-\infty }^{\infty }\frac{s(\tau )}{t-\tau }d\tau$$
where $h(t)=\frac{1}{\pi t}$.

WVD is a kind of time–frequency distribution of Cohen’s class, and its formula is:(2)$$C(t,f)=\int_{-\infty }^{\infty }\int_{-\infty }^{\infty }\int_{-\infty }^{\infty }a(u+\frac{\tau }{2})a(u-\frac{\tau }{2})\varphi (\tau ,v)e^{-j2\pi (tv+\tau f-uv)}dudvd\tau$$
where ϕ(τ,v) is the kernel function. When the kernel function ϕ(τ,v)=1, Formula (2) becomes:(3)$$W_{a}(t,f)=\int_{-\infty }^{\infty }a(u+\frac{\tau }{2})a(u-\frac{\tau }{2})e^{-j2\pi \tau f}d\tau$$

Formula (3) is the Wigner–Ville distribution.

#### 2.1.2. Smoothed Pseudo-Wigner–Ville Distribution

When the WVD is applied to multi-component signals, different signal components are prone to cross-action in the calculation. This phenomenon will lead to blurred signal characteristics and seriously affect the analysis results. Therefore, a method is needed to effectively suppress the cross terms.

The influence of cross terms on the analysis results can be effectively reduced by adding kernel functions to the WVD. A typical algorithm for this method is the smoothed pseudo-Wigner–Ville distribution (SPWVD), as shown in Formula (4):(4)$$SPW_{a}(t,f)=\int_{-\infty }^{\infty }a(t-u+\frac{\tau }{2})a(t-u-\frac{\tau }{2})g(u)h(\tau )e^{-j2\pi \tau f}d\tau$$
where g(u)h(τ) is the added kernel function, g(u) is the time domain window function, h(τ) is the frequency domain window function, and g(0)=h(0)=1.

### 2.2. Convolutional Neural Network

The convolutional neural network is mainly composed of a convolution layer, a pooling layer, and a fully connected layer, which is essentially a feature extractor of the input signal. The input signal of the network is generally a multi-layer matrix. The signal features are extracted through the convolutional layer, and the pooling layer reduces the number of model parameters. Finally, the fully connected layer realizes the mapping from the feature layer to the output and realizes the function of classification.

#### 2.2.1. Convolution Layer

The convolution layer is the core of the convolutional neural network. Assuming the convolution kernel of I×J in the convolution layer is *k* and the weight in the convolution kernel is $\omega_{i,j}$, then, the calculation formula of the convolution layer is:(5)$$f^{l}=\sum_{i=0,j=0}^{i=I,j=J}\omega_{i,j}*F^{l-1}+b$$

In the formula, *f* is an element in the new feature matrix, *l* is the number of feature layers, $F^{l-1}$ is the previous feature matrix or input matrix, and *b* is the bias.

Let the size of the output matrix after the convolution calculation be W×H×D; then, the size of the output matrix can be calculated by the Formulas (6)–(8):(6)$$W=\frac{W_{0}-Filter+2P}{S}+1$$
(7)$$H=\frac{H_{0}-Filter+2P}{S}+1$$
(8)D=N
where $W_{0}$ and $H_{0}$ are the input matrix size, *Filter* is the convolution kernel size, *P* is the number of filled zeros, *S* is the convolution kernel sliding step size, and *N* is the number of convolution kernels. When the convolution kernel slides to the boundary of the input matrix, some elements will not be calculated, so the feature matrix is filled by zero filling. The number of zero filling, *P*, is:(9)$$P=\frac{Filter-S}{2}$$

#### 2.2.2. Pooling Layer

The feature layer obtained by the convolution layer usually has a large amount of data and cannot be directly used for classification. Therefore, it is necessary to down-sample through the pooling layer to effectively reduce the number of network parameters and control the overfitting to a certain extent. The expressions when using the maximum and average pooling operations are shown in Formulas (10) and (11):(10)$$f^{l}=\max(p^{l-1})$$
(11)$$f^{l}=\mathrm{mean}(p^{l-1})$$
where *p* is the pooling layer filter.

The pooling layer usually uses a 2 × 2 size filter to slide on the feature layer in a step of 2, and the maximum value in each sliding window is retained to form a new feature layer for the next operation. The size of the characteristic matrix after pooling is shown as follows in (12)–(14):(12)$$W=\frac{W_{0}-Filter}{S}+1$$
(13)$$H=\frac{H_{0}-Filter}{S}+1$$
(14)$$D=D_{0}$$

It can be seen from the above formula that there is no zero-padding operation in the pooling layer, and the depth of the feature layer is unchanged before and after the pooling operation.

#### 2.2.3. Fully Connected Layer

After feature extraction, multiple features need to be connected for the classification. The fully connected layer is a plurality of W×H×D convolution kernels, and the size is completely the same as the previous feature layer, turning the feature layer into one-dimensional data. The dimension of the data entirely depends on the number of categories. After the calculation of the fully connected layer is completed, the output is sent to the SoftMax classifier to complete the classification. The expression of the SoftMax classifier is:(15)$$S_{i}=\frac{e^{a_{i}}}{\sum_{j=1}^{J}e^{a_{j}}}$$
where $a_{i}$ represents the *i*-th data of the output of the fully connected layer, *J* represents the total output of the fully connected layer, and $S_{i}$ represents the possibility of the corresponding category.

### 2.3. Activation Function: Leaky RelU

Different from the most commonly used RelU activation function, an activation function called Leaky RelU is introduced in the model, which is expressed as follows [18]:(16)$$y=\left\{\begin{array}{l} x\\ \alpha x \end{array}\right.\begin{array}{c} ,\\ , \end{array}\begin{array}{c} x>0\\ x\leq 0 \end{array}$$
where α is a constant close to 0. When the input is positive, it is equivalent to the RelU activation function, and the input remains unchanged. When the input is negative, because a is a given small constant, some values of the negative axis are retained, so that the information of the negative axis is not lost, and the gradient can propagate normally.

## 3. Proposed Shuffle-BiLSTM Model

According to the functional characteristics of group convolution and channel shuffle, combined with the BiLSTM structure, this paper constructed the Shuffle-BiLSTM network. Its network architecture is shown in Figure 1.

The network model is mainly composed of three parts: the shuffle unit module, BiLSTM module, and the vibration state monitoring module. Firstly, the 2D time–frequency spectrum of the three-way acceleration signal and the sound pressure signal extracted by SPWVD was adjusted to 0–255 grayscale images. A 256 × 256 × 3 matrix was formed by these images to realize the fusion of the two sensor signals in time–frequency domain features. Next, after the initial convolution pool and other operations, the shuffle unit was entered. The shuffle unit is mainly composed of the above group convolution and the channel shuffle, plus the necessary data batch normalization, pooling, and leaky RelU layers, and it imitates the residual network to increase the short-circuit mechanism, which avoids the gradient explosion problem in the training process, to a certain extent. Considering that the group convolution operation will reduce the accuracy of the network, multiple sets of shuffling units were added to increase the network depth. Then, after the shuffle unit, a BiLSTM layer was added to further extract the time series features of the data and adaptively filter the corresponding types of implicit features in the data. Finally, the learned features were fed back to the classification layer, composed of the fully connected layer and the SoftMax classifier, to identify and output the vibration state of the boring bar.

### 3.1. Shuffle Unit

The shuffle structure was originally a lightweight network structure model proposed by Xiangyu Zhang et al. [19]. This structure greatly reduced the parameters of the deep learning model by introducing group convolution and channel shuffle, and effectively improved the calculation speed and accuracy of the model.

#### 3.1.1. Group Convolution

According to the above content, the convolution kernel in the ordinary convolution layer will output all the information of the feature layer to the next feature layer through the convolution operation, as shown in Figure 2a. The parameter quantity in the convolution operation is:(17)$$Q=Filter\times N\times D_{0}$$
where *D*_0_ is the number of layers of the input matrix, that is, the depth of the input layer.

The grouping convolution is different from the ordinary convolution layer. It groups the input layer and then uses different group convolutions for the calculation. The corresponding group convolution kernel is only convoluted with the corresponding input layer, as shown in Figure 2b. The parameters of group convolution are:(18)$$Q_{G}=Filter\times N\times \frac{D_{0}}{G}$$
where *G* is the number of groups. From the Formula (17), Formula (18) can be seen as: $Q_{G}=\frac{Q}{G}$. The operation of group convolution can greatly reduce the number of parameters and improve the calculation speed of the model.

#### 3.1.2. Channel Shuffle

Although the computational cost can be significantly reduced by grouping convolution, this method makes the output of each group only come from a part of the input layer, as shown in Figure 3. Obviously, groups are isolated from each other, and there is no information flow, which will reduce the learning ability of the model.

To solve the problem of grouping convolution, the channel shuffle method was introduced. After the channel shuffle method was applied to the grouping convolution operation, the channels of each group were further grouped according to the total number of groups, and then mixed with each other to ensure that each large group can have the characteristics of the other groups and be used as the input layer of the next convolution operation. The specific implementation method is shown in Figure 4. Let the number of feature layer groupings be G and the total depth be D. After combining them into (G, D), it was transposed and re-leveled to achieve channel shuffling. It can be seen from the diagram that the mixed channel after channel mixing avoided the separation and isolation between the above channels, to a certain extent.

### 3.2. Bidirectional Long Short-Term Memory Network

Long short-term memory (LSTM) recognizes, stores, and forgets features through the ‘gate’ mechanism. LSTM can use this memory-like feature to filter and save the implicit features of the input data, identify and save the features associated with the current state, discard redundant features, and repeatedly update the memory as the data are continuously input.

LSTM has three gate functions, namely the input gate, forgetting gate, and the output gate. The forgetting gate determines which memory information needs to be modified and passed to the next step. The input gate is responsible for integrating the previous memory and the new input, and the output gate will pass the filtered memory information down. The mathematical expression is as follows [20]:(19)$$\left\{\begin{array}{l} F_{t}=\sigma (W_{f}\left[h_{t-1},x_{i}\right]+b_{f}\\ I_{t}=\sigma (W_{i}\left[h_{t-1},x_{i}\right]+b_{i})\\ {\tilde{c}}_{t}=\tanh(W_{c}\left[h_{t-1},x_{t}\right]+b_{c})\\ c_{t}=f_{t}\cdot c_{t-1}+I_{t}\cdot \tilde{c}\\ \sigma_{t}=\sigma (W_{u}\left[s_{t-1},x_{t}\right]+bu)\\ h_{t}=o_{t}\cdot \tanh\left(c_{i}\right) \end{array}\right.$$
where $I_{t}$, $F_{t}$, $o_{t}$, and $c_{t}$, respectively, represent the corresponding input gate, forgetting gate, output gate, and cell unit state at time t. $W_{f}$, $W_{i}$, $W_{c}$, and $W_{u}$ are the weights of each door. $b_{f}$, $b_{i}$, $b_{c}$, and $b_{u}$ are the offsets of the respective doors. $\sigma \left(\cdot \right)$ stands for the sigmoid activation function, $x_{t}$ is the input information at time t, and $h_{t-1}$ and $h_{t}$ are the hidden layer information at time t−1 and t, respectively. $c_{t-1}$ is the cell state information at the time t−1 and ${\tilde{c}}_{t}$ is the candidate cell state.

Since the feature transfer of LSTM has a direction, a single transfer direction may not be able to fully extract the implicit features in the data. Therefore, bidirectional long short-term memory (BiLSTM) was introduced, and the data feature sequence was bidirectionally extracted and combined by the combination of forward LSTM and backward LSTM to better capture the data sequence features and achieve accurate classification of the state.

The overall structure of BiLSTM is shown in Figure 5.

### 3.3. Vibration State Monitoring Module

The vibration state monitoring module is responsible for outputting the vibration state of the monitored boring bar, which is mainly composed of a full connection layer and a classifier, as shown in Figure 1. The full connection layer performed weighted regression on the advanced features learned from the shuffling unit module and the BiLSTM module, and finally connected the classifier to identify the three vibration states of the boring bar and output the corresponding labels.

## 4. Experiment

### 4.1. Experimental Device and Data Acquisition

To establish the boring experiment dataset and provide enough data for the intelligent perception of the boring bar vibration state for training, boring experiments are needed. The experiment changed the boring cutting speed, feed rate, cutting depth, and boring bar overhang. The acceleration sensor was installed at the front end of the boring bar to collect the acceleration signal during the cutting process, the sound sensor was used to collect the noise pressure of the machining noise, and the surface roughness and image were recorded. The CNC lathe used in the experiment was the CKA6150 (Shenyang Machine Tool Group Co., Ltd., Shenyang, China), the cutting workpiece was 45 steel, the outer diameter was 180 mm, the inner diameter was 140 mm, the total length was 200 mm, the diameter of the boring bar was 60 mm, and the maximum overhang length was 600 mm. The blade model was CCMT060604, and the workpiece rotated during cutting, the boring bar feeds, there was no cutting fluid, and the maximum feed depth was 30 mm. The data acquisition hardware used the NI PXIE-1092 (National Instruments (NI) Inc., Shanghai, China) data acquisition box, integrated PXIE-4492 vibration and sound acquisition modules, and the software used the LabVIEW (National Instruments (NI) Inc.) data acquisition system. Through the trial cutting experiment, the collected signals were analyzed, and it was found that the highest frequency of the vibration signal and the sound pressure signal was not higher than 10 KHz. Under the premise of computer memory and computing resources, according to the Shannon sampling theorem, the sampling frequency was set to 40 kHz. The diagram and photos of the experimental site are shown in Figure 6, and Figure 7 shows the data acquisition interface.

The NI PXIE-1092 data acquisition box was used for the experimental data acquisition. The chassis was equipped with a high-bandwidth backplane and 82 W power supply and cooling functions. It provides timing and synchronization options, a built-in constant temperature crystal oscillator, and supports PXIE modules. The acquisition box integrates PXIE-4492 vibration and sound acquisition modules, supports a maximum sampling frequency of 204.8 kS/S, and is equipped with a SHB4X-8BNC connection cable. The type of acceleration sensor used was the Donghua Test 1A314E IEPE (Jiangsu Donghua Testing Technology Co., Ltd., Taizhou, China), with a maximum range of 50 g. The sound sensor was the CRY SOUND CRY333 (CRY SOUND, Hangzhou, China) free-field measurement microphone, which includes a CRY507 IEPE (CRY SOUND) preamplifier with a maximum range of 146 dB.

The detailed cutting parameters and the overhang of the boring bar are shown in Table 1. A total of 192 sets of cutting experiments were carried out.

### 4.2. Feature Extraction

#### 4.2.1. Signal Denoising

Considering the influence of the experimental environment, the collected signal contained noise, which will seriously affect the data analysis, resulting in a low model monitoring accuracy. Therefore, under the premise of ensuring the characteristics of the original signal, multiple sets of wavelet packet threshold denoising were performed on the original signal. The noise reduction process is shown in Figure 8.

Through research, it was found that for acceleration signals, the signal SNR using the unbiased likelihood estimation threshold, the hard threshold function, the coif5 basis function, and three-layer wavelet decomposition was the highest, at 8.13, and the RMSE was the lowest, at 1.30. This parameter had the best noise reduction effect and is most suitable for acceleration signals. For the acoustic emission signal, when the unbiased likelihood estimation threshold, the hard threshold function, and the coif5 basis function were selected, the SNR of the signal was the highest, at 11.1, and the RMSE was the lowest, at 0.01. Considering that increasing the number of decomposition layers increases the amount of calculation, the least three-layer wavelet decomposition was selected among the three decomposition layers with similar noise reduction effects.

#### 4.2.2. SPWVD Feature Extraction

The acceleration and sound pressure signal with the best noise reduction effect were selected as the target data, and SPWVD was used to obtain the three-dimensional time–frequency spectrum of the signal. Due to the limitation of space, the cutting speed was 200 m/min, the feed speed was 0.2 mm/rot, and the extension was 540 mm. When the cutting depth increased from 0.1 mm to 0.4 mm, the time–frequency spectrum of the calculated tangential acceleration signal was as shown in Figure 9.

The time–frequency relationship of the signal can be roughly seen from the 3D time–frequency spectrum. In order to more intuitively express this relationship, the 3D image was projected downward along the *z*-axis (amplitude) direction to obtain a 2D time–frequency spectrum, as shown in Figure 10.

This kind of time–frequency spectrum can reflect the frequency domain distribution of signal components in time series, and it is an effective signal time–frequency analysis method. However, the analysis of this image is difficult to directly quantify by humans, such as time domain and frequency domain features, and the readability is poor. Therefore, it can be identified and analyzed by deep learning algorithms that are good at image recognition. The image was transformed into a three-channel 0–255 gray value matrix, which can be identified by a deep learning model.

### 4.3. Model Training and Comparison Experiment Settings

#### 4.3.1. Model Training

A total of 192 cutting experiments were randomly grouped, 70% of the data (135 groups) was used to train the model, and 30% (57 groups) was used to test the model.

To obtain a higher monitoring accuracy, a set of optimal model training parameters were selected through the experimental methods. The minibatch size was set to 64, maxepochs to 1000, initial learn rate to 0.1, each iteration was performed 100 times, the learning rate multiplication coefficient was 0.1, and using SGDM optimization, the momentum was 0.9.

The final recognition accuracy of the training set was used as the training effect standard. 

#### 4.3.2. Comparison Experiment Settings

To verify the accuracy and superiority of the shuffle-BiLSTM model proposed in this study, five classic deep network models: convolutional neural network (CNN), Shuffle NET, Res NET, Dense NET, and Inception NET, were selected. On the premise of using the same dataset, five models were trained for comparison, and the model used in this paper was trained using a single vibration signal. The performance was compared with the models and methods used in this paper. The training and test methods of the above models were the same as those used in this paper.

### 4.4. Results’ Analysis and Discussion

#### 4.4.1. Experimental Result Analysis

Figure 11 shows the training and test results of the proposed model. Figure 12 shows the training and test results of the comparison model. Table 2 is the performance parameter table of the proposed model and the comparison model.

As shown in Figure 11a, after 756 iterations, the model training accuracy reached 100%, indicating that all groups in the training set were able to achieve correct classification.

Only using the training set training model to improve the training accuracy cannot complete the construction of the model. It is also necessary to input the test set into the model and continuously adjust the model parameters to make the test accuracy high enough. The remaining 57 sets of experimental data were input into the model as a test set for testing. The confusion matrix of the test results is shown in Figure 11b. Among them, 24 groups of the 26 groups of the stable cutting state were accurately identified, with an accuracy rate of 92.3%, 13 groups of the 14 groups of the transition state were accurately identified, with an accuracy rate of 92.9%, 15 groups of the 17 groups of the violent vibration were accurately identified, with an accuracy rate of 88.2%, and the overall classification accuracy was 91.2%. The ‘0’ label in the graph indicates a stable cutting state, the ‘1’ label indicates a transition state, and the ‘2’ label indicates a violent vibration state.

In the above model, the original convolutional neural network structure was the simplest: the number of network layers was only 14, the number of parameters was only 0.8 M, the amount of calculation was the least, and the recognition speed was the fastest. However, due to its depth limitation, the ability to extract the implicit information in the data was insufficient, and the learning speed was slower than the model and the Shuffle NET. After 650 training iterations, it finally converged.

Shuffle NET is the basic version of the network used in this paper. The network uses a large number of group convolution kernels, which significantly reduces the complexity of the network. Even if the number of network layers reached 172, the number of parameters was only 1.4 M, and the calculation amount was very small. After 250 training iterations, the convergence was completed, and the learning ability was excellent. However, due to the influence of group convolution on the accuracy of the model, the test accuracy was slightly lower, which was 86.0%.

Res NET, Dense NET, and Inception NET are all more complex deep learning networks. By observing the test accuracy of these three networks it can be found that their test accuracy was ideal. The overall test accuracy of Dense NET and Inception NET exceeded the network model used in this article. Res NET was limited by the network depth, and the overall classification accuracy was slightly lower than the other two models, but the parameter quantity was only half of the other two models. Although the test results of the three networks were good, they all had more than 10 M parameters. Dense NET and Inception NET even reached 20 M, which is 10 times the proposed model. Due to its complex network structure and more internal feature layers, convergence can be achieved with fewer iterations, but the higher number of parameters makes the calculation time of the network model very long and the recognition speed slow.

When using this model to sense the vibration state of a single-signal input, the speed of network extraction and learning features was accelerated due to the reduction of input data, and convergence was achieved after 445 iterations. However, due to the limitation of only one input signal, the final classification accuracy was not high, which also shows the effectiveness of the multi-signal fusion input method used in this paper.

#### 4.4.2. Discussion

Based on the performance comparison of several network models, the shuffle-BiLSTM NET used in this paper has a lightweight network depth, low computational complexity, and enough recognition accuracy to meet the requirements. The comprehensive performance was the best, which is suitable for production and processing scenarios.

The good comprehensive performance of shuffle-BiLSTM NET may come from the following aspects. The method of extracting local features by group convolution reduced the calculation amount of the model and the operation time. A channel shuffling layer was added under the convolution layer to realize cross-extraction and learning of features, which avoided the limitations brought by local feature extraction of group convolution. The BiLSTM structure was added to enhance the network’s memory ability, so that the network had a better recognition ability for similar states. Although the training time of the network increased, the recognition accuracy effectively improved.

According to Table 2, the shuffle-BiLSTM network proposed in this study had a small amount of calculation, a fast running speed, and the monitoring accuracy could also meet the monitoring requirements, and the comprehensive performance was optimal. However, compared with large-scale deep models, such as Dense NET and Inception NET (long running time), its monitoring accuracy was slightly lower. Compared with the ordinary CNN model, shuffle-BiLSTM had a higher monitoring accuracy, but the running time was slightly longer.

## 5. Conclusions and Future Works

In the process of deep-hole-boring, the vibration state of the boring bar is difficult to monitor. To solve this problem, this study proposed an intelligent monitoring technology of the boring bar’s vibration state based on data acquisition, signal processing, and deep learning technology. Applying the proposed technology to the boring monitoring system, the vibration state of the boring bar can be perceived in real-time. Operators can adjust the processing parameters according to the perception results to improve the efficiency and accuracy of processing. Through a large number of experimental studies and comparison with some traditional depth models, the effectiveness and superiority of the model were verified. The main conclusions are as follows:

(1) The secondary time–frequency representation method with the kernel function (SPWVD) was used to process the experimental data. The original data were transformed into a two-dimensional time–frequency spectrum, and this was identified by the deep network model. The deeper features were extracted, and the effective classification of the vibration state was realized.

(2) The group convolution method was used to extract some features of the input layer, and the group convolution layer was rearranged by channel shuffling. This reduced the amount of calculation, shortened the calculation time of the model, and improved the real-time monitoring, while avoiding the cognitive limitations of the model. BiLSTM was used to extract and screen the data memory characteristics, which enhanced the memory ability of the network and realized the accurate classification of the boring bar’s vibration state.

(3) The cutting experiments of different vibration states of the boring bar were designed, and 192 groups of cutting experiments were carried out by changing different experimental parameters. The vibration and sound pressure data in the experiment were collected and used as the original data of the vibration state perception. The deep network model was trained and tested, and the test classification accuracy of the model used in this paper was 91.2% when the parameter quantity was only 1.9 M. A variety of typical deep network models and a single-signal input model were added for performance comparison testing. The test results showed the advantages of the models and methods used in this paper.

This study provides a better choice for civil and military enterprises involved in deep-hole-boring. To better guide industrial production, in the future research work, the following potential research directions can be further explored: how to accurately identify the vibration state of the boring bar under variable working conditions and design an appropriate deep transfer learning model to deal with small samples or incomplete datasets.

## Figures and Tables

**Figure 1 sensors-23-06123-f001:**
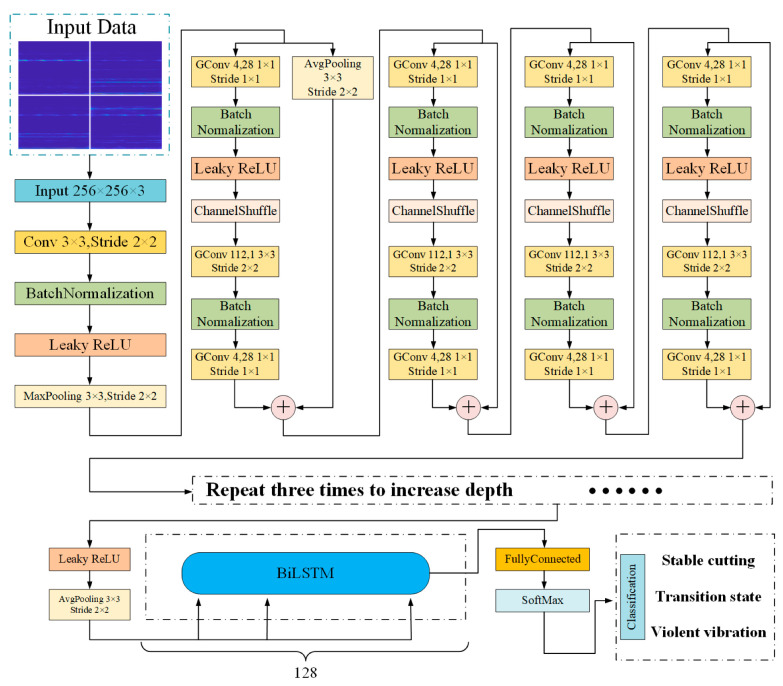
Structure of the network.

**Figure 2 sensors-23-06123-f002:**
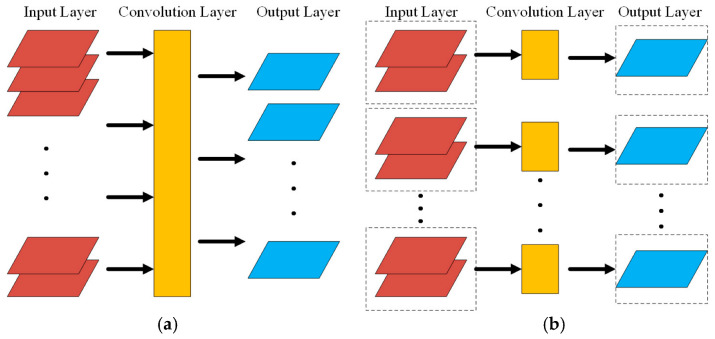
Ordinary convolution and group convolution. (**a**) Ordinary convolution. (**b**) Group convolution.

**Figure 3 sensors-23-06123-f003:**
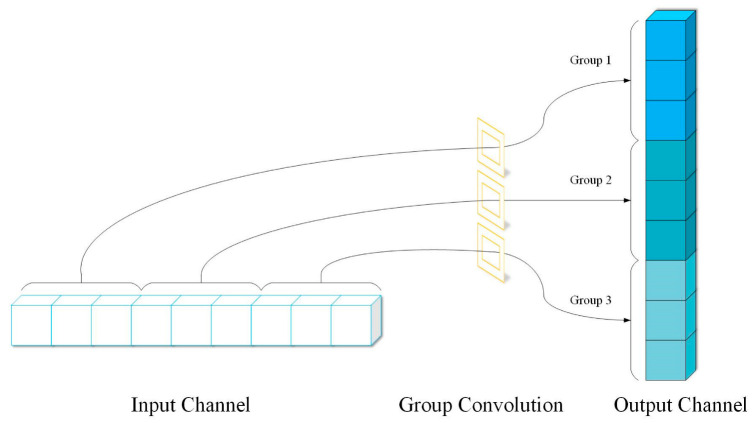
Channel separation phenomenon.

**Figure 4 sensors-23-06123-f004:**
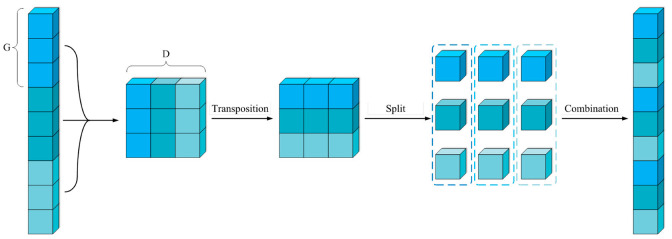
Channel shuffle.

**Figure 5 sensors-23-06123-f005:**
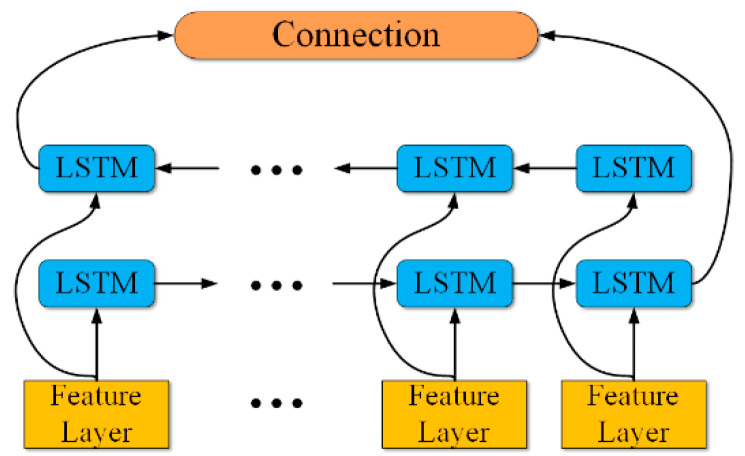
Structure of BiLSTM.

**Figure 6 sensors-23-06123-f006:**
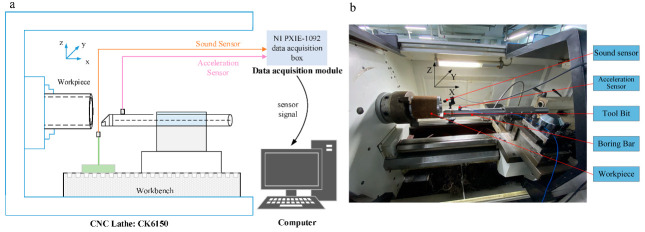
Experimental site: (**a**) illustration and (**b**) actual set-up.

**Figure 7 sensors-23-06123-f007:**
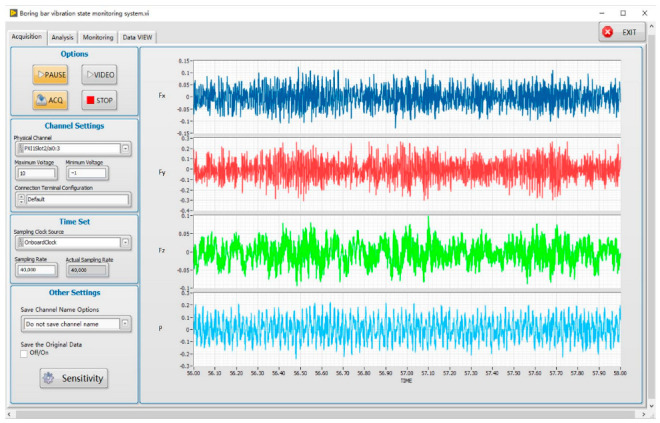
Data acquisition interface.

**Figure 8 sensors-23-06123-f008:**
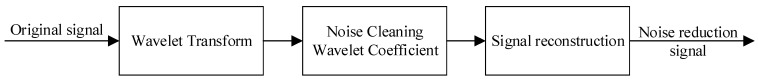
Wavelet threshold denoising.

**Figure 9 sensors-23-06123-f009:**
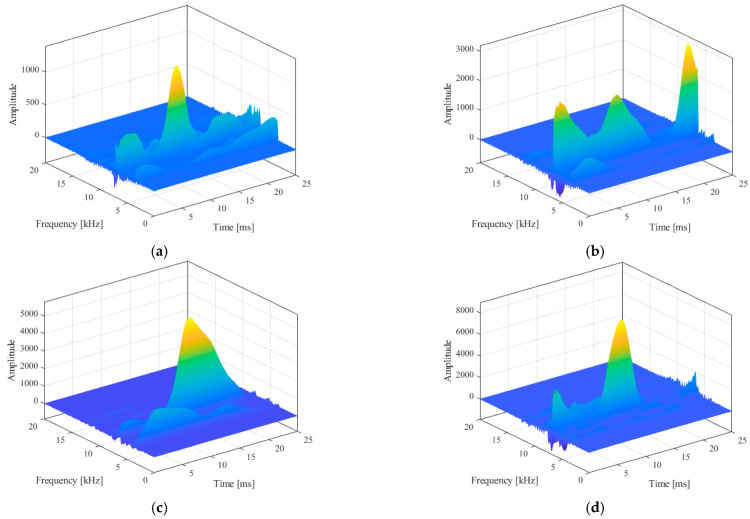
Vibration acceleration signal 3D time–frequency spectrum diagram. (**a**) a_p_ = 0.1 mm, (**b**) a_p_ = 0.2 mm, (**c**) a_p_ = 0.3 mm, and (**d**) a_p_ = 0.4 mm.

**Figure 10 sensors-23-06123-f010:**
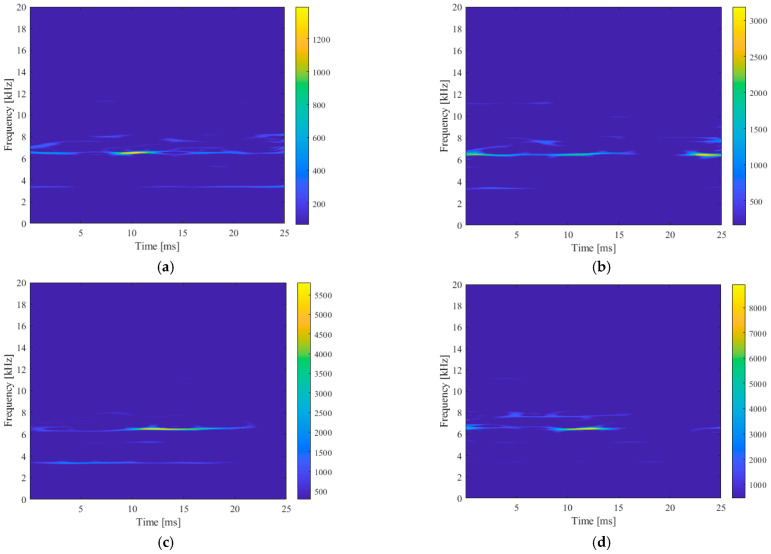
Vibration acceleration signal 2D time–frequency spectrum diagram. (**a**) a_p_ = 0.1 mm, (**b**) a_p_ = 0.2 mm, (**c**) a_p_ = 0.3 mm, and (**d**) a_p_ = 0.4 mm.

**Figure 11 sensors-23-06123-f011:**
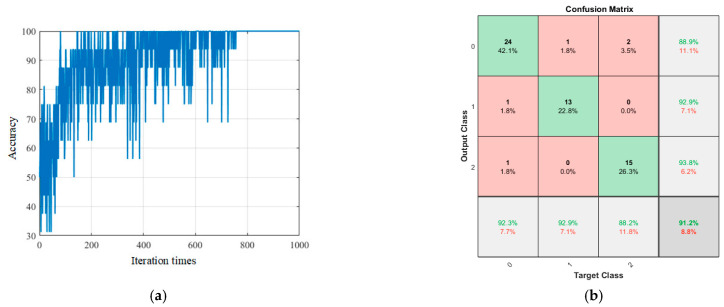
Training and test results of the proposed model. (**a**) Training results and (**b**) test results.

**Figure 12 sensors-23-06123-f012:**
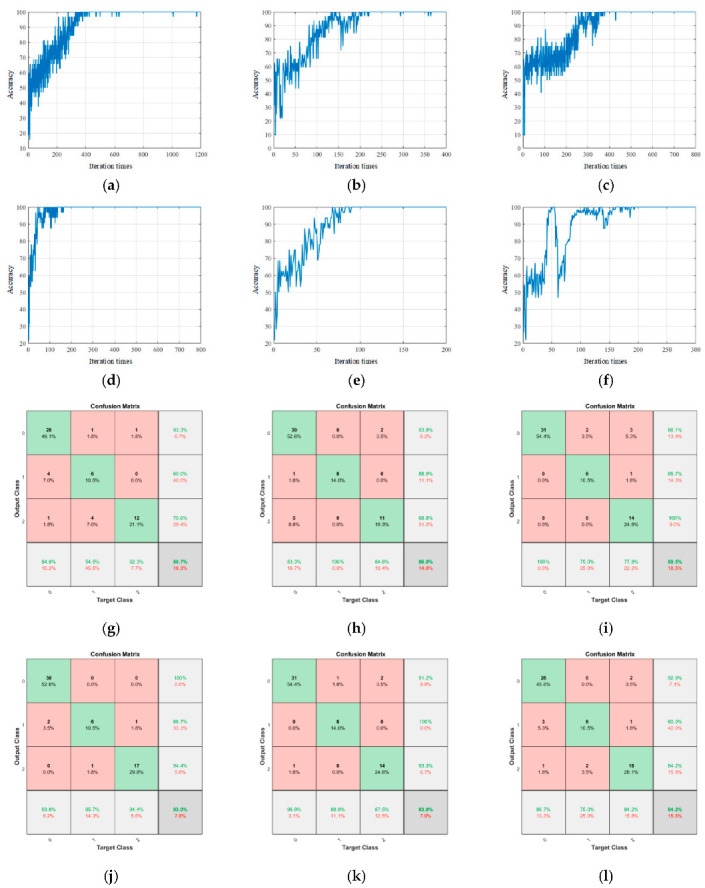
Training and test results of the comparison experiment. (**a**) CNN, (**b**) Shuffle NET, (**c**) Res NET, (**d**) Dense NET, (**e**) Inception NET, (**f**) single signal, (**g**) CNN, (**h**) Shuffle NET, (**i**) Res NET, (**j**) Dense NET, (**k**) Inception NET, and (**l**) single signal.

**Table 1 sensors-23-06123-t001:** Experimental parameters.

No.	Cutting Speed (m/min)	Feed Rate (mm/r)	Cutting Depth (mm)	Overhang (mm)
1	100	0.1	0.1	420
2	200	0.2	0.2	480
3	300	0.3	0.3	540
4	400	0.4	0.4	-

**Table 2 sensors-23-06123-t002:** Model performance comparison.

Model Name	Iteration Times	Test Accuracy				Network Layers Number	Parameters Number
		Stable	Transition	Violent	Overall		
Shuffle-BiLSTM NET	756	92.3%	92.9%	88.2%	91.2%	176	1.9 M
CNN	650	84.8%	54.5%	92.3%	80.7%	14	0.8 M
Shuffle NET	250	83.3%	100%	84.6%	86.0%	172	1.4 M
Res NET	430	100%	75%	77.8%	89.5%	71	11.7 M
Dense NET	170	93.8%	85.7%	94.4%	93.0%	708	20.0 M
Inception NET	90	96.9%	88.9%	87.5%	93.0%	315	23.9 M
Single signal	445	86.7%	75.0%	84.2%	84.2%	176	1.9 M

## Data Availability

Not applicable.

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
