# Peer review of "Research on Intelligent Monitoring of Boring Bar Vibration State Based on Shuffle-BiLSTM"

_sensors, 2023, doi:10.3390/s23136123_

Round 1

Reviewer 1 Report

1. The Introduction include all references and provide sufficient background

2. All references are relevant to the research

3 The Research Design is appropriate

4. The  paper is interest to readers and sounds scientifically well

Adwice: probably for more information ,  write a liitlle about future research  using the same  methodology or equipment

Reviewer 2 Report

The authors used an acceleration sensor to measure the vibrations, in my opinion; they need to add a section on acceleration sensors and strengthen it with new work such as:

- Appropriate Choice of Damping Rate and Frequency Margin for Improvement of the Piezoelectric Sensor Measurement Accuracy

-Improvement of the vibratory diagnostic method by evolution of the piezoelectric sensor performances

- The use of mechanical sensitivity model to enhance capacitive sensor characteristics

- Piezoresistive accelerometer mathematical model development with experimental validation

You need to correct spelling mistakes.

Reviewer 3 Report

The paper needs the following significant improvement before publication:

First, Extensive English proofreading is required. The use of personal pronouns should be avoided in technical papers.

Please check if the formulae require citations from an appropriate source.

The novelty/objective of the paper should be highlighted in more detail. Hence, more focus should be given to the last paragraph of the Introduction.

The authors need to revise the format of the paper. For example, it should be generally based on Introduction, materials and methods or experimental setup/procedure, results and discussion, etc. Hence, after the introduction section, in this paper, the organization of the paper is confusing and sometimes too lengthy to follow.

Most importantly, a separate discussion section is required, which includes a critical analysis based on the results.

The conclusions section requires future recommendations regarding industrial applications.

Extensive English proofreading is required.

Reviewer 4 Report

Figure 6 – unclear, better to be replaced or completed by a schematic diagram of the experimental layout

Line 126: “with sampling frequency of 40k.” – perhaps 40 KHz (or kilo samples /second). Also, please justify why this value was chosen (Shannon theorem or anything else)

Line 310: “feed rate of 0.2mm/r” – consider 0.2 mm/rot (rotation) or 0.2 mm/rev (revolution)

It is not clear if the signals from sensors were pre-processed (filtered) before being analyzed by SPWVD. Please elaborate.

The practical usefulness of the proposed approach is quite unclear. Please add more regarding that in the conclusion section.

English language acceptable.

Round 2

Reviewer 3 Report

The authors have revised the manuscript. The paper is now accepted for publication. Thank you.

Minor English proofreading is required. 

Reviewer 4 Report

The authors have addressed all issues raised by the reviewer, providing solid and precise solutions for each. Thus, I consider that the paper can now be published in the journal as such.